# Perhexiline: Old Drug, New Tricks? A Summary of Its Anti-Cancer Effects

**DOI:** 10.3390/molecules28083624

**Published:** 2023-04-21

**Authors:** Bimala Dhakal, Yoko Tomita, Paul Drew, Timothy Price, Guy Maddern, Eric Smith, Kevin Fenix

**Affiliations:** 1Discipline of Surgery, Adelaide Medical School, Faculty of Health and Medical Sciences, The University of Adelaide, Adelaide, SA 5005, Australia; bimala.dhakal@adelaide.edu.au (B.D.); paul.drew@adelaide.edu.au (P.D.); guy.maddern@adelaide.edu.au (G.M.); 2Basil Hetzel Institute for Translational Health Research, The Queen Elizabeth Hospital, Adelaide, SA 5011, Australia; yoko.tomita@sa.gov.au (Y.T.); timothy.price@sa.gov.au (T.P.); 3Medical Oncology, The Queen Elizabeth Hospital, Adelaide, SA 5011, Australia; 4Adelaide Medical School, Faculty of Health and Medical Sciences, The University of Adelaide, Adelaide, SA 5005, Australia

**Keywords:** perhexiline, repurposing, anti-cancer, cancer metabolism, cardiotoxicity

## Abstract

Cancer metabolic plasticity, including changes in fatty acid metabolism utilisation, is now widely appreciated as a key driver for cancer cell growth, survival and malignancy. Hence, cancer metabolic pathways have been the focus of much recent drug development. Perhexiline is a prophylactic antianginal drug known to act by inhibiting carnitine palmitoyltransferase 1 (CPT1) and 2 (CPT2), mitochondrial enzymes critical for fatty acid metabolism. In this review, we discuss the growing evidence that perhexiline has potent anti-cancer properties when tested as a monotherapy or in combination with traditional chemotherapeutics. We review the CPT1/2 dependent and independent mechanisms of its anti-cancer activities. Finally, we speculate on the clinical feasibility and utility of repurposing perhexiline as an anti-cancer agent, its limitations including known side effects and its potential added benefit of limiting cardiotoxicity induced by other chemotherapeutics.

## 1. Introduction

Cancer cells have high metabolic demands and many altered metabolic pathways, diverting nutrients toward the anabolic processes required to sustain the production of ATP and macromolecules required for growth and survival in harsh tissue microenvironments. In particular, various aspects of fatty acid uptake, synthesis, modification and degradation are altered in cancer cells during tumour development and progression (extensively reviewed in [1,2,3]). Alterations in fatty acid β-oxidation (FAO) influence cancer cell proliferation, metastasis, stemness, survival and drug resistance [1]. Therefore, key enzymes or regulators of FAO are potential targets for cancer therapy.

The carnitine system plays an essential role in FAO, being critical for the transport of long-chain fatty acids across the mitochondrial membrane to generate energy. The carnitine system consists of four components: carnitine palmitoyltransferase 1 (CPT1; 3 isoforms CPT1A, CPT1B and CPT1C) and 2 (CPT2), the carnitine-acylcarnitine translocase and the carnitine acetyltransferase. As key regulators of FAO, these components have been implicated in cancer progression/survival and are often overexpressed in different cancer types [4]. Targeting the carnitine system has been proposed as a cancer treatment strategy [5], but nothing so far has translated into clinical practice.

Multiple CPT inhibitors have been developed, including perhexiline and etomoxir (reviewed in [6]). The anti-anginal drug perhexiline can inhibit CPT1 and CPT2 [7,8]. Recent preclinical studies suggest that perhexiline possesses significant anti-tumour activity. It has been used as a monotherapy, or in combination therapy with other anti-cancer drugs to enhance their therapeutic efficacy or reduce drug resistance. Although the activities of perhexiline have been widely investigated, there remains uncertainty about the underlying mechanisms of action. This review summarises the potential mechanisms of action of perhexiline based on reported studies and discusses the potential of re-purposing this drug as a treatment for cancer.

## 2. Overview of Perhexiline

Perhexiline maleate (prescribed as Pexsig or Pexid) was originally developed as an anti-anginal drug in the 1970s (reviewed in [6]). It was an effective treatment for angina, but reports of severe hepatoxicity and neurotoxicity in a subset of patients resulted in reduced global usage by the late 1980s [9,10]. Nevertheless, perhexiline continues to be prescribed in several countries, including Australia and New Zealand, where it is approved for treating patients with refractory angina pectoris or patients with angina in whom other therapies are contraindicated [11,12]. Other reported clinically beneficial effects include enhancing myocardial energetics in heart failure [13] and hypertrophic cardiomyopathy [14].

Perhexiline, 2-(2,2-dicyclohexylethyl) piperidine, is a small, amphiphilic molecule with a –CH-CH2 carbon chain backbone, two saturated cyclohexane rings and a piperidine ring (Figure 1). It is a chiral molecule due to asymmetry of the second carbon of the piperidine ring, and the clinical formulation of perhexiline is a racemic 1:1 mixture of (+)-perhexiline and (−)-perhexiline enantiomers. Perhexiline is metabolised by the polymorphic enzyme cytochrome P450 2D6 (CYP2D6). Variable expression of and/or mutations in CYP2D6 can result in highly variable clearance rates, with distinct phenotypes termed ultrarapid, extensive, intermediate and poor metabolisers [15]. Pharmacokinetic studies suggest that the perhexiline enantiomers display stereoselective metabolism, with the clearance rate of (−)-perhexiline greater than (+)-perhexiline [16,17,18].

The clinical benefits of perhexiline are derived from its ability to inhibit the mitochondrial enzymes CPT1 and CPT2 [7,8]. As part of the carnitine system, CPT1 and CPT2 are involved in the translocation of fatty acids across the mitochondrial membranes into mitochondrial matrix, where they undergo FAO (β-oxidation). The oxidation process generates acetyl-CoA, nicotinamide adenine dinucleotide (NADH) and flavin adenine dinucleotide (FADH2). Acetyl-CoA enters the tricarboxylic acid (TCA) cycle generating more NADH and FADH2, which are co-enzymes in the electron transport chain, and these intermediates are used to generate cellular adenosine triphosphate (ATP). In the cardiac setting, inhibition of the CPTs by perhexiline shifts myocardial metabolism from principally fatty acid toward a greater carbohydrate metabolism, maintaining myocardial production of ATP but requiring lower oxygen consumption [19].

Despite its unique mechanism of action and proven efficacy, the clinical use of perhexiline is limited owing to its narrow therapeutic index and variable pharmacokinetics [10,20,21]. These factors mean that for long-term clinical use, plasma concentrations of perhexiline must be monitored and maintained within a therapeutic range of 0.15 to 0.60 mg/L (equivalent to ~0.5 to 2 µmol/L) [22]. With monitoring, the risk of serious toxicity associated with long-term perhexiline dosing can be minimised, without abolition of the antianginal effects [11,22]. Further, perhexiline toxicity takes several months to develop. Therefore, much higher doses than normally prescribed can be tolerated if perhexiline is administered in cycles, provided there are sufficient breaks between the cycles [11,22].

## 3. Perhexiline Anti-Cancer Studies and Proposed Mechanisms

Multiple studies have demonstrated that perhexiline by itself inhibits the growth of various cancer cell lines in vitro at micromolar concentrations (Table 1). Rodriguez-Enriquez et al. reported the half-maximal inhibitory concentrations (IC50) for perhexiline ranged between 3 and 22 µmol/L for a variety of cancer cell lines, including from the breast, cervix, colon and lung [23]. Ren et al. reported that the IC50 for multiple breast cancer cell lines was between 2 and 6 µmol/L [24]. We recently reported that the IC50 for a panel of colon cancer cell lines was around 4 µmol/L [25]. These IC50 ranges are similar to the recommended maximum therapeutic plasma concentration of perhexiline of around 2 µmol/L [21]. Small animal studies have demonstrated that perhexiline concentrates further within tissues [26]. Thus, there are anti-tumour effects when the plasma concentration is within the therapeutic range. However, the relationship between the plasma and tumour perhexiline concentrations has not been explored.

Perhexiline appears to be moderately selective for cancer cells. The growth of colon cancer cell lines was inhibited by lower concentrations of perhexiline than human foreskin fibroblasts, with a selectivity index of approximately three [25]. Likewise, normal liver organoids were more tolerant of perhexiline than colorectal cancer (CRC) patient-derived organoids (PDOs) [25]. Metastatic HeLa cells were three times more sensitive than fibroblast lines (3T3 and CCD-25Lu) and non-metastatic cancer cell lines [23]. Wang et al. reported that gastric cancer (HGC27, MGC803) and colon cancer (HCT116, DLD-1) cell lines were more sensitive to perhexiline than cell lines derived from normal gastric (GES-1) and colonic (CCD 841) epithelium [32]. Perhexiline induced apoptosis in the gastrointestinal cancer cell lines at 10 µmol/L, while cells derived from normal epithelium were resistant up to 20 µmol/L. Furthermore, they observed that expression of *CPT1B* and *CPT2* was generally higher in the cancer cells, raising the possibility that increased CPT expression levels may contribute to perhexiline sensitivity. Liu et al. demonstrated that patient-derived chronic lymphocytic leukemia (CLL) cells were more sensitive to perhexiline than normal lymphocytes and bone marrow stromal cells [30]. They postulated that the turnover rate of cardiolipin may be higher in CLL than normal cells due to elevated intrinsic levels of oxidative stress and higher expression of cardiolipin degradation enzymes such as the phospholipase iPLA2 [46,47,48]. The high cardiolipin turnover would render CLL cells dependent on the transport of fatty acids into the mitochondria for cardiolipin synthesis, and susceptible to CPT inhibition. Perhexiline selectivity may be dependent on the relative balance between nutrient availability and requirement. Transient exposure of immortalised mouse embryo fibroblasts to perhexiline was not toxic in nutrient-rich conditions, but lead to rapid apoptosis in starvation conditions by mechanisms involving inhibition of the nutrient sensor, mammalian target of rapamycin complex 1 (mTORC1) [29]. Interestingly, these perhexiline mediated anti-cancer effects have been reported to involve multiple cellular processes (Figure 1).

### 3.1. Perhexiline Activates the Intrinsic Apoptotic Pathway

Multiple studies have reported that the growth inhibition induced in cancer cells by micromolar concentrations of perhexiline involved the induction of apoptosis [25,30,32,33,34,38,43,44,49]. Apoptosis is a highly regulated form of programmed cell death [50], and the intrinsic (non-receptor mediated) apoptotic pathway is initiated by intracellular cues, such as oxidative stress [51]. Studies in gastrointestinal cancer cells revealed that perhexiline-induced apoptosis was associated with decreases in FAO and NADPH production, and a significant increase in the level of reactive oxygen species (ROS) [32]. These changes are known to contribute to mitochondrial membrane permeability [51]. Indeed, perhexiline induced the loss of mitochondrial membrane integrity, release of cytochrome c, activation of caspase-3 and caspase-9, and cleavage of poly (ADP-ribose) polymerase 1 (PARP-1) [25,30,33,43].

In contrast, little is known about the effects of nanomolar concentrations of perhexiline on apoptosis. Brown et al. demonstrated that the high fat environment of non-alcoholic fatty liver disease (NAFLD) promoted CD4^+^ T-cell apoptosis through peroxisome proliferator-activated receptor alpha (PPARalpha)-mediated upregulation of CPT gene expression [36]. Increased CPT expression promoted mitochondrial uptake of the long chain fatty acid linoleic acid, resulting in elevated mitochondrial ROS levels and cell apoptosis. Nanomolar concentrations of perhexiline, 312.5 and 625 nmol/L in murine and human CD4^+^ T-cells, respectively, inhibited the linoleic acid-induced apoptosis in vitro. Furthermore, in a diet-induced transgenic mouse model of NAFLD, perhexiline treatment inhibited the spontaneous development of hepatocellular carcinoma (HCC). This was associated with a reduction in apoptotic events in intrahepatic CD4^+^ T-cells, which are known to play critical roles in tumour cell surveillance and elimination. However, it is not known if cancer cells respond in a similar way to nanomolar concentrations of perhexiline.

### 3.2. Perhexiline Promotes Incomplete AMP-Activated Protein Kinase (AMPK) Activated Autophagy

Autophagy is a tightly regulated and stress-induced catabolic pathway in which cellular targets are engulfed by autophagosomes and delivered to lysosomes for degradation into metabolic substrates [52,53]. Autophagy will be triggered by energy starvation or starvation of nutrients such as amino acids which act through the 5′-adenosine monophosphate (AMP)-activated protein kinase (AMPK) signalling pathway or mammalian target of rapamycin (mTOR) complex 1 (mTORC1), respectively [54]. Changes in intracellular AMP/ATP levels promote phosphorylation and activation of AMPK (pAMPK). Once activated, pAMPK switches off energy-consuming processes, and switches on ATP-generating mechanisms.

Studies in cardiac tissues have shown that perhexiline (1–10 µmol/L) activates the phosphorylation of AMPK, but only in the absence of glucose [55]. Meanwhile, Xu et al. demonstrated that concomitant inhibition of FAO with perhexiline, glycolysis and oxidative phosphorylation (OXPHOS) resulted in AMPK phosphorylation in hepatocellular carcinoma cell lines in vitro and in vivo [37]. Interestingly, Bagli et al. identified perhexiline from a screen of chemicals that were capable of inducing autophagy and inhibiting mTORC1 signalling in an MCF-7 breast cancer cell line maintained in nutrient-rich conditions [29]. Autophagy induction and inhibition of mTORC1 signalling was rapidly reversible upon perhexiline withdrawal, unlike with the mTOR inhibitor rapamycin. The authors suggested that perhexiline most likely inhibited mTORC1 signalling by acting on upstream regulatory pathways. Itkonen et al. observed that perhexiline induced incomplete autophagy in LNCaP prostate cancer cells leading to cell death [43].

### 3.3. Perhexiline Improves Chemotherapy Efficacy

The first description of the anti-cancer potential of perhexiline was published by Ramu et al. in 1984 [27]. Building on the observation that doxorubicin-resistant and sensitive leukaemia cell lines could be distinguished based on their cell lipid membrane composition [56], they investigated if perhexiline combined with doxorubicin could overcome resistance. They observed that perhexiline re-sensitised doxorubicin-resistant murine B-cell lymphoma cells (P388/ADR) to doxorubicin treatment. Similar results were observed in doxorubicin-resistant breast cancer cells (MCF-7/ADR) [28]. The increased sensitivity to doxorubicin in MCF-7/ADR was accompanied by an increase in intracellular doxorubicin accumulation [28], raising the possibility that perhexiline reduced the activity of ATP-binding cassette (ABC) transporters, which are involved in drug efflux and chemotherapeutic resistance [57].

Perhexiline improved the efficacy of platinum-based antineoplastics oxaliplatin and cisplatin in colon, gastric [32], neuroblastoma [39] and epithelial ovarian cancer [33]. Wang et al. reported that oxaliplatin treatment significantly increased expression of *CPT1B* and *CPT2* in gastric (HGC27 and MGC803) and colon (HCT116 and DLD-1) cancer cells [32]. In comparison, cisplatin and 5-fluorouracil (5-FU) slightly increased the expression of these enzymes. Expression of *CPT1B* and *CPT2* was elevated in an oxaliplatin-resistant derivative of HCT116 (HCT116-OxaR). High expression of CPT1B and CPT2, determined by immunohistochemistry, was associated with poor response to oxaliplatin-based therapy (FOLFOX or XELOX) in advanced colorectal cancer patients [32]. Perhexiline (10 to 20 µmol/L) inhibited proliferation and induced apoptosis in the gastrointestinal cancer cell lines, but not in cell lines derived from normal gastric (GES-1) or colonic (CCD841) epithelium. Furthermore, the HCT116-OxaR cells exhibited greater sensitivity to perhexiline than HCT116. The combination of perhexiline and oxaliplatin greatly inhibited tumour growth of several gastrointestinal cancer cell lines and patient-derived xenografts (PDXs) [32]. The therapeutic response in gastric cancer PDXs correlated with the basal levels of CPT1 and CPT2 expression. Together, these findings raised the possibility that perhexiline could be used to overcome oxaliplatin resistance in gastrointestinal cancers.

Zhu et al. reported that perhexiline and cisplatin synergistically inhibited epithelial ovarian cancer cell lines bearing deletions of the NK2 homeobox 8 (*NKX2-8*) gene [33]. The *NKX2-8* gene is a homeobox-containing developmental regulator that is down-regulated in multiple cancers and associated with disease progression [58,59,60,61]. Deletions in *NKX2-8* were associated with reprogramming of fatty acid metabolism, including increased *CPT1A* and *CPT2* expression, and increased chemoresistance. The combination of perhexiline and cisplatin markedly reduced the growth of an epithelial ovarian cancer (EOC) cell line xenograft bearing a deletion in the NK2 homeobox 8 (*NKX2-8^+^*^/−^) gene. This feature may also contribute to the cancer specificity observed in other studies mentioned above.

Upregulation of HER3 is a major mechanism underlying resistance to epithelial growth factor receptor (EGFR) and HER2 tyrosine kinase inhibitors, including lapatinib [62,63,64]. Treatment of HER3+ breast cancer cells with lapatinib induces a transient decrease in activation of HER3 and downstream Akt signalling [24,63]. Ren et al. observed that the addition of perhexiline to lapatinib prolonged inhibition of HER3 signalling [24]. Furthermore, the combination of perhexiline and lapatinib synergistically inhibited proliferation of some breast cancer cell lines.

Treatment with luminespib (NVP-AUY922), a second-generation heat shock protein 90 (HSP90) inhibitor, increased the abundance of proteins involved in OXPHOS and fatty acid metabolism in prostate cancer patient-derived explants, and increased mitochondrial mass and expression of genes associated with fatty acid metabolism processes, including *CPT1A*, in prostate cancer cell lines [44]. The combination of luminespib and perhexiline synergistically decreased viability of prostate cancer cell lines and increased the efficacy in explants by inducing cell cycle arrest and apoptosis, and attenuated the heat shock response, a known mediator of HSP90 treatment resistance.

Vella et al. [39] demonstrated that relatively low concentrations of perhexiline (0.01 and 1 µmol/L) increased the expression of neuroblastoma differentiation marker 29 (NDM29), a non-coding RNA known to induce differentiation and restrict the tumorigenic potential of neuroblastoma cells [65]. The increased NDM29 expression was associated with a reduction in ABC transporter (*ABCA1*, *ABCA12*) and solute carrier (*SLC7A11*) expression. A non-cytostatic concentration of perhexiline (0.01 µmol/L) enhanced the cytotoxic effect of cisplatin synergistically in SH-SY5Y neuroblastoma cells. In a mouse neuroblastoma xenograft model, co-administration of cisplatin and perhexiline was more effective than either therapeutic alone at inhibiting tumour growth and prolonging survival, demonstrating the potential for dose reduction of a chemotherapeutic that is commonly associated with toxicity in patients [39].

### 3.4. Perhexiline Improves Anti-Androgen Therapy Efficacy

Prostate cancer cells differ from those of many other cancers in that they predominantly utilise fatty acid rather than glucose metabolism, and fatty acid metabolism is an androgen-regulated process in prostate cancer cells [66]. Studies have shown that improved anti-cancer efficacy can be achieved by co-targeting androgen signalling and FAO, which may have clinical implications especially in metastatic castrate-resistant prostate cancer patients [42,43]. Flaig et al. reported that CPT1A expression was increased in prostate cancer compared to benign tissue. Decreased CPT1A expression was associated with decreased Akt content and activation. Since lipid oxidation is stimulated by androgens, the effects of combining CPT1A inhibition and anti-androgen therapy was evaluated. The combination therapy resulted in increased androgen receptor action and increased sensitivity to the anti-androgen enzalutamide. The combination of FAO inhibitors (perhexiline, etomoxir or ranolazine) with enzalutamide produced robust growth inhibition in prostate cancer cell models including enzalutamide resistant LNCaP and mouse TRAMPC1 cells [42]. Itkonen et al. demonstrated that perhexiline promoted intracellular accumulation of lipid, inhibited proliferation and induced incomplete autophagy and apoptosis in LNCaP prostate cancer cell lines [43]. Furthermore, perhexiline combined with the anti-androgens abiraterone (ABI) or enzalutamide (MDV-3100) almost completely blocked proliferation.

### 3.5. Perhexiline as Part of a Metabolic Inhibitor Strategy for Cancer

Many cancer cells prefer glycolysis for their energetic needs (Warburg effect). The hexokinase family consisting of four genes (HK1-4) are rate-limiting enzymes in glycolysis. While cancer cells can express more than one of these isoenzymes, the majority only use HK2. Xu et al. recognised that a HK1^−^HK2^+^ cancer subset existed among a wide variety of cancer types, including multiple myeloma (MM) [38] and hepatocellular carcinoma (HCC) [37]. They demonstrated that perhexiline synergised with the glycolysis inhibitor 2-fluoro-deoxy-D-glucose (FDG) and the OXPHOS inhibitor diphenyleneiodonium (DPI) to significantly decrease viability of liver cancer cells in vitro and in mouse xenografts [37]. Since glycolysis is an important metabolic process utilised by normal tissues, Xu et al. further refined this triple combination by replacing FDG with shRNA knockdown of HK2 expression [38]. The triple combination strategy was shown to be more effective for treating HK2-positive tumours than the dual therapy of HK2-knockdown combined with OXPHOS inhibition.

Overexpression of 3-phosphoglycerate dehydrogenase (PHGDH), the rate-limiting enzyme in the biosynthesis of serine from glucose, has been observed in many cancer types and has been linked to poor patient outcomes including chemotherapy resistance, shorter progression-free survival, increased rates of metastasis and poorer overall survival (reviewed in [67]). Rathore et al. observed that PHGDH inhibition in osteosarcoma cell lines attenuated cellular proliferation without causing cell death, prompting an extensive metabolic analysis to characterise pro-survival mechanisms [40]. Metabolomic and lipidomic profiling of the cellular response to PHGDH inhibition revealed the accumulation of unsaturated lipids, branched chain amino acids and methionine cycle intermediates, leading to activation of pro-survival mTORC1 signalling. The combination of perhexiline with a PHGDH small molecule inhibitor (NCT-503) resulted in significant synergistic cell death in vitro and in vivo, providing preclinical justification for a dual metabolism-based combination therapy for PHGDH-high cancers.

## 4. Perhexiline: More than Just CPT Inhibition

While perhexiline is widely considered to act by inhibiting CPT and FAO, a number of reports suggest that its anti-cancer effects may be mediated by CPT-independent pathways.

### 4.1. PI3K/Akt/mTOR

The phosphoinositide 3-kinase (PI3K)-Akt-mammalian target of rapamycin (mTOR) signalling pathway coordinates the uptake and utilisation of multiple nutrients, including lipids, glucose, glutamine and nucleotides, facilitating the enhanced growth and proliferation of cancer cells. The pathway is one of the most frequently altered in human cancers. Several studies have demonstrated that perhexiline suppresses the PI3K/Akt/mTOR pathway [24,33,40,44].

PI3K transduces upstream signals from receptor tyrosine kinases to generate critical lipid second messengers that activate downstream signalling effectors such as Akt and mTOR. The conserved serine/threonine-protein kinase, mTOR, belongs to the PI3K family of protein kinases and constitutes the catalytic component of two distinct multiprotein complexes: mTOR complex 1 (mTORC1) and mTOR complex 2 (mTORC2) [68]. The mTORC1 complex contains regulatory-associated protein of mTOR (raptor), and the mTORC2 complex contains the rapamycin-insensitive companion of mTOR (rictor). The mTORC1 catalyses the phosphorylation of S6 kinase β-1 (S6K) and initiation factor 4E binding protein 1 (4E-BP1). In contrast, mTORC2 initiates phosphorylation of Akt and protein kinase C (PKC), thereby regulating nutrient metabolism, protein synthesis, growth factor signalling, cell growth and migration [69,70].

Balgi et al. identified that perhexiline inhibited mTORC1 signalling, as evidenced by decreases in S6K and 4E-BP1 phosphorylation, and induced autophagy in the breast cancer cell line MCF-7 maintained in nutrient-rich conditions [29]. In contrast, perhexiline did not inhibit mTORC2, suggesting that it did not inhibit mTOR catalytic activity, but rather inhibited signalling to mTORC1. Similarly, Rathore et al. demonstrated that perhexiline treatment of an osteosarcoma cell line (NOS1) significantly decreased levels of total mTOR and total RSP6 [40]. Nassar et al. demonstrated that perhexiline induced modest inhibition of ERK and Akt phosphorylation in C4-2B prostate cancer cells [44].

Inhibition of the PI3K/Akt/mTOR pathway has been observed with other CPT inhibitors [71]. Combination therapy with etomoxir (an irreversible inhibitor of CPT1A), and orlistat (an irreversible inhibitor of lipases and fatty acid synthase) resulted in a synergistic decrease in viability of prostate cancer cell lines (LNCaP and VCaP) and patient-derived benign and prostate cancer cells. These effects were associated with decreased mTOR signalling, decreased androgen receptor expression, and increased apoptosis. Knockdown of *CPT1A* expression in LNCaP cells decreased oxidation of the fatty acid palmitic acid (C16:0), increased sensitivity to etomoxir, inactivated Akt and activated apoptosis.

### 4.2. ErbB3 (HER3)

The ErbB family of proteins consist of four structurally related receptor tyrosine kinases: ErbB1 (HER1, EGFR), ErbB2 (HER2, Neu), ErbB3 (HER3) and ErbB4 (HER4). Excessive ErbB signalling is associated with the development of various solid tumours and is associated with poor patient outcomes. Heterodimerisation of HER3 with HER2 activates the PI3K/Akt/mTOR pathway [72], and HER3 knockout impairs the ability of HER2 to induce tumour formation [73].

Ren et al. demonstrated that perhexiline inhibited the activation of HER3 and the proliferation of HER3+ breast cancer cell lines in vitro and in xenografts [24]. The perhexiline treatment induced rapid HER3 internalisation and a reduction in phosphorylated HER3 (pHER3), an active form for HER3 signalling. In contrast, expression of other ErbB family members (EGFR and HER2) were unaffected by perhexiline treatment. Etomoxir had no effect on HER3 localisation, providing limited evidence that perhexiline may operate through mechanisms independent of CPT inhibition. Furthermore, perhexiline inhibited phosphorylation of Akt and ERK1/2, downstream effectors of HER3 signalling. Similarly, Zhu et al. demonstrated that perhexiline inhibited phosphorylation of Akt and other downstream HER3 activation markers in *NKX2-8*-deleted (*NKX2-8*^+/−^) CAOV3 and OVCAR3 epithelial ovarian cancer (EOC) cells [33].

### 4.3. FYN

Tyrosine-protein kinase Fyn (FYN) is a membrane-associated non-receptor tyrosine kinase belonging to the Src family of kinases. It is aberrantly expressed in various cancers contributing to multifaceted signalling that regulates aspects of tumour progression including cellular differentiation, proliferation, antiapoptotic activity, increased migration and motility [74,75,76,77,78]. FYN negatively regulates AMPK activity, thereby promoting cell migration and invasion through AMPK/mTOR-mediated signalling [78]. Furthermore, FYN plays a functional role in modulating redox stress by negatively regulating NAPD oxidases that inhibit production of ROS [79].

Kant et al. investigated the effects of perhexiline on FYN in glioblastoma [34]. Perhexiline induced concentration-dependent cytotoxicity in undifferentiated glioblastoma stem cells (PN19 and MES83), and these were more sensitive than differentiated cell lines (T98G and U251). Induced differentiation of PN19 and MES83 increased resistance to perhexiline. However, in contrast to etomoxir, they observed that 5 µmol/L perhexiline failed to alter the oxygen consumption rate in MES83 and T98G cells and did not modulate intracellular lipid dynamics in MES83, leading to the conclusion that the anti-tumour effects of perhexiline were independent of CPT and FAO inhibition in these cells. To identify alternative perhexiline targets, they used SwissTargetPrediction in silico analysis [80], and identified several high probability molecular targets of perhexiline, including FYN, EGFR and family AG protein-coupled receptors (muscarinic acetylcholine receptor M1, M3-5). Surprisingly, CPT1 was not a reported high-probability target. Expression of FYN was high in the perhexiline-sensitive and low in the resistant cell line T98G, unlike the other top targets. FYN expression was significantly higher in glioblastoma, particularly in the proneural subtype, than in normal brain. Perhexiline treatment of glioblastoma cell lines (PN19, MES83 and U251) inhibited FYN activation, as determined by a time-dependent increase in FYN phosphorylation.

### 4.4. HES1

Hairy and enhancer of split-1 (HES1) is one of seven members of the Hes gene family (HES1-7) that encode basic helix-loop-helix transcription factors which suppress transcription. HES1 has a central role in NOTCH1-induced leukaemia suggesting that abrogation of HES1 activity in leukemia lymphoblasts could be exploited therapeutically. To identify potential small molecule inhibitors of HES1, Schnell et al. interrogated the Connectivity Map [81], a large collection of genome-wide transcriptional expression data derived from cell lines treated with bioactive small molecules, for compounds with transcriptional signatures that overlapped with that induced by HES1 deletion in NOTCH1-induced T-cell acute lymphoblastic leukaemia (T-ALL) [45]. Perhexiline was identified as a potential therapeutic agent for T-ALL due to its ability to elicit a gene expression signature resembling that induced by HES1 deletion in NOTCH1-induced T-ALL. Perhexiline downregulated HES1 expression in CUTLL1 T-ALL cells in vitro. Notably, perhexiline treatment resulted in a significant anti-tumour response and extended survival in mice bearing NOTCH1-induced T-ALL, without significant detrimental effects on the hematopoietic system.

## 5. Preclinical Studies of Perhexiline-Mediated Tumour Clearance

The in vivo anti-tumour potential of perhexiline has been investigated in mouse studies for numerous cancer types, including breast [24], chronic lymphocytic leukaemia [30], colorectal [32], gastric [32], glioblastoma [34], liver [36,37], multiple myeloma [38], neuroblastoma [39], osteosarcoma [40], ovarian [33] and T-cell acute lymphoblastic leukaemia [45] (Table 2). Most studies evaluated xenografts of human cancer cell lines, and occasionally patient-derived tumour tissue. A spontaneously arising cancer in transgenic immunocompetent mice has been studied [36]. Perhexiline was administered as either monotherapy or combination therapy. The perhexiline dosage varied between studies, with doses from 1 to 400 mg/kg, delivery by oral gavage or intraperitoneal injection and assorted dosing regimens. The perhexiline dosages appeared to be well tolerated, with minimal toxicity reported. Perhexiline accumulation was detected in the brain, demonstrating that perhexiline can cross the blood brain barrier [34]. However, the perhexiline blood concentration was not measured in any of these studies. Despite promising anti-tumour findings in numerous preclinical in vitro and mouse studies, there are currently no registered clinical trials evaluating the efficacy of perhexiline (Pexsig or Pexid) for the treatment of cancer in humans.

## 6. Perhexiline Modulates Tumour-Infiltrating Immune Cells

Most studies investigating the anti-cancer effects of perhexiline have focused solely on its direct effect on cancer cells with pre-clinical models using immunocompromised hosts (Table 2). There is now mounting evidence that perhexiline treatment can also affect tumour immune infiltrates. Several groups have studied the effect of perhexiline on macrophages [82,83,84,85,86]. Macrophages are heterogenous immune cells commonly found in tumours and can dictate both positive and negative outcomes depending on their phenotype and function. Tumour-associated macrophages are broadly classified into pro-inflammatory M1-like and anti-inflammatory M2-like macrophages. Infiltration of M2-like macrophages is associated with increased tumour immune suppression, angiogenesis, cancer outgrowth and metastasis [87,88]. In contrast, M1-like macrophages release type 1 pro-inflammatory cytokines that drive anti-tumour immune responses. While M1-like macrophages have an increased reliance on glycolysis, M2-like macrophages have an elevated requirement for FAO and OXPHOS [89,90]; thus, influencing macrophage phenotypes by targeting metabolic pathways is viewed as a promising new cancer treatment [82]. Perhexiline has been shown to inhibit M2-like and promote M1-like polarization, and can even drive repolarization of established M2-like macrophages to M1-like macrophages most likely by inhibiting FAO [82,83]. Interestingly, perhexiline can also suppress macrophage-mediated inflammation by downregulating IL-1β [83,84,85]. How perhexiline affects tumour-associated macrophages in vivo has yet to be explored. Additionally, in an immunocompetent model of obesity-associated breast cancer, perhexiline treatment led to significant tumour inhibition that corresponded with infiltration of interferon-gamma and granzyme b producing CD8+ T cells [86]. In this study, obesity-induced STAT3 activation on CD8+ T cells led to a metabolic shift to FAO that produced immune-suppressed CD8+ T cells, which was reversed by perhexiline treatment. However, FAO inhibition on CD8+ T cells alone did not completely replicate the extent of tumour inhibition, suggesting that perhexiline may act through multiple targets to mediate its anti-tumour effects [86]. Together, these results suggest that perhexiline has the potential to modulate tumour-infiltrating immune cells, which may further enhance its anti-tumour efficacy in vivo.

## 7. Clinical Feasibility of Perhexiline as an Anti-Cancer Agent

Combining perhexiline with platinum compounds in cancer therapeutics is an area of research that should be expedited. Platinum compounds such as carboplatin, cisplatin and oxaliplatin exhibit their anti-cancer activity by interfering with normal DNA functions and are integral to systemic treatment of diverse ranges of solid and haematological cancers [91]. Platinum compounds are one of the most widely used anti-cancer therapeutics. Given the reported synergy between perhexiline and platinum agents [32,39], perhexiline has the potential to increase the efficacy of existing platinum-containing cancer therapy regimens for various cancers. Perhexiline has been additionally shown to sensitise epithelial ovarian cancer cells to cisplatin by counteracting *NKX2-8* deletion-induced platinum resistance [33]. Platinum resistance is a well-recognised prognostic factor for ovarian cancer and its presence predicts worse survival outcomes with reduced systemic therapy options [92]. Combining perhexiline with platinum compounds may overcome the development of platinum resistance in ovarian cancer management and improve its clinical outlook.

Combining perhexiline with cancer therapeutics that share the same site of action is another proposed approach to further exploring perhexiline’s clinical utility. PI3K, mTOR and HER3 receptor, previously discussed as potential biological targets of perhexiline, are recognised sites of action for existing clinically available cancer therapeutics. Alpelisib has been FDA-approved in combination with fulvestrant for treatment of hormone receptor-positive HER2-negative breast cancer harbouring PIK3CA mutations [93]. The mTOR inhibitor everolimus is a systemic treatment option for hormone receptor-positive HER2 receptor-negative advanced breast cancer when co-administered with exemestane, in addition to renal cell carcinoma and pancreatic neuroendocrine tumours [94]. Pertuzumab, a humanised monoclonal antibody which inhibits dimerisation of HER2 and HER3 receptors, is used in conjunction with trastuzumab and chemotherapy for HER2-positive breast cancer in both locally advanced and metastatic settings [95,96]. Adjunctive use of perhexiline may increase the inhibition of biological target of interest and result in enhanced overall anti-cancer efficacy. The observation in animal studies that perhexiline concentrates within visceral organs [26] and crosses the blood-brain barrier [34] further makes this drug a favourable cancer therapeutic candidate. It may be a particular benefit for tumours in the central nervous system where the blood-brain barrier, by restricting the effective delivery of many cancer drugs, makes treatment of primary brain cancer and central nervous system metastasis challenging [97].

Perhexiline may also have clinical utility in cancer management, independent of direct actions on cancer cells, due to its cardioprotective effects. There are an increasing number of cancer therapeutics the use of which are associated with a risk of cardiovascular adverse effects. Anthracyclines and anti-Her2 agents, such as trastuzumab, are known to carry a risk of cardiomyopathy and congestive heart failure and fluoropyrimidines, including 5-fluorouracil and capecitabine, are associated with a risk of cardiac ischaemia [98]. Other drugs with less common cardiovascular toxicities include alkylating agents such as cyclophosphamide and ifosfamide, which are linked to neurohumoral activation-induced heart failure, and vinca alkaloids such vincristine and vinblastine with a risk of cardiac ischaemia and congestive heart failure. Anthracyclines, commonly used in the treatment of breast cancer and sarcoma, are the cancer therapeutics most frequently implicated with cardiotoxicity. They cause clinically overt cardiotoxicity in 6% of patients receiving the drugs with an additional 18% of patients experiencing subclinical cardiotoxicity [99]. The ability of perhexiline to improve cardiac energetics may underly its benefit in alleviating symptoms of heart failure and angina [6,100]. The co-treatment with perhexiline in patients prescribed cancer therapeutics associated with cardiovascular toxicities might reduce the incidence of clinically significant heart failure and cardiac ischaemia.

## 8. Conclusions

There is growing appreciation of the anti-cancer properties of perhexiline, with most studies being published in the last 10 years (Table 1 and Table 2). Perhexiline inhibits tumour growth at concentrations that are achievable and safe for clinical use as a short-term cancer therapy. While perhexiline was initially described as acting through CPT inhibition, it is now clear that perhexiline has many other actions. More research is required to determine if these other effects are specific primary targets of perhexiline or if these could be downstream effects of CPT inhibition, cell damage or cell death resulting from perhexiline treatment. Furthermore, the effect of perhexiline on the tumour microenvironment should be investigated. The potential of perhexiline in combination therapy to enhance the response to conventional cytotoxic drugs holds great potential and warrants further research. With its positive effects on the tumour immune response, perhexiline should also be tested for synergies with immunotherapies such as immune checkpoint inhibitors. The safety profile of perhexiline is relatively well understood, suggesting that clinical translation should be feasible. Perhexiline appears to be a good candidate for repurposing for the clinical management of cancer patients.

## Figures and Tables

**Figure 1 molecules-28-03624-f001:**
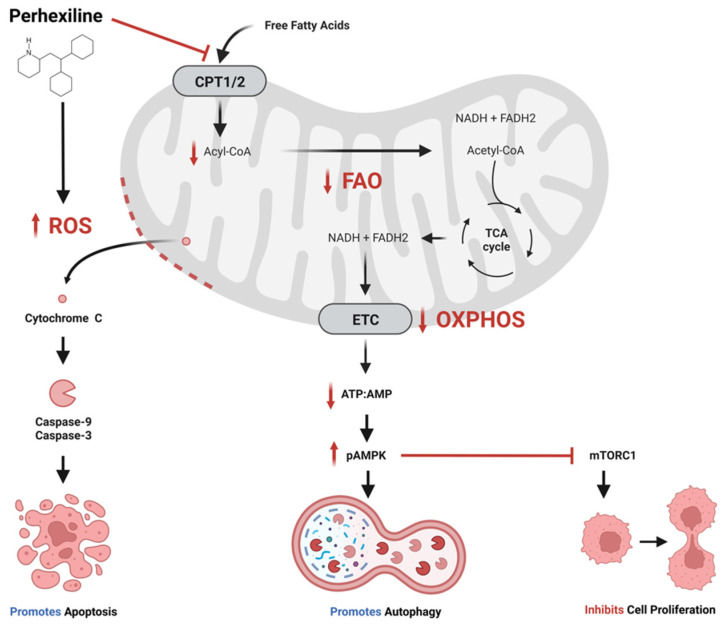
**Proposed anti-cancer effects of perhexiline.** Perhexiline inhibition of carnitine palmitoyltransferase 1 (CPT1) and CPT2 restricts the entry of free fatty acids into the mitochondrial matrix, thereby inhibiting fatty acid oxidation (FAO). This may limit the production of the electron transport chain (ETC) co-enzymes nicotinamide adenine dinucleotide (NADH) and flavin adenine dinucleotide (FADH2), which would inhibit oxidative phosphorylation (OXPHOS) and the generation of adenosine triphosphate (ATP). A reduction in ratio of ATP to adenosine monophosphate (AMP) activates AMP-activated protein kinase (AMPK) by phosphorylation (pAMPK). pAMPK triggers autophagy, and inhibits cell proliferation by inhibiting mammalian target of rapamycin complex 1 (mTORC1). Additionally, perhexiline increases the level of reactive oxygen species (ROS), which compromises mitochondrial membrane integrity, leading to the release cytochrome c and activation caspases that initiate apoptosis. Created with BioRender.com.

**Table 1 molecules-28-03624-t001:** Studies summarising the effects of perhexiline on various cancer cells in vitro.

Study	Cancer	Cell Lines	Key Findings
Ramu et al. (1984) [27]	BCL	P388 P388/ADR	Re-sensitised DOX-resistant P388/ADR to DOX.
Foster et al. (1988) [28]	BRCA	MCF-7 MCF-7/ADR	Re-sensitised DOX-resistant MCF-7/ADR to DOX. Increased intracellular DOX accumulation.
Balgi et al. (2009) [29]	BRCA	MCF-7	Induced autophagy. Inhibited mTOR signalling.
Ren et al. (2015) [24]	BRCA	MDA-MB-468	Inhibited growth.
		SK-BR-3	Induced HER3 internalisation and degradation.
		AU565	Synergistic with lapatinib.
		BT474	Overcame lapatinib induced resistance.
Rodriguez-Enriquez et al. (2015) [23]	BRCA CAC CRC Lung Fibroblast	MDA-MB-231 MDA-MB-468 MCF-7 HeLa COLO205 A-549 3T3 CCD-25Lu	Inhibited growth.
Liu et al. (2016) [30]	CLL	Primary CLL, normal lymphocytes	Inhibited growth. Induced apoptosis. Did not decrease oxygen consumption.
Batra & Alenfall (1991) [31]	CRC	HT-29	Inhibited growth.
Dhakal et al. (2022) [25]	CRC Fibroblast	COLO205 HCT116 HT-29 SW480 SW620 PDO HFF	Inhibited growth. Induced apoptosis.
Wang et al. (2020) [32]	CRC GC	HCT116 DLD-1 HGC27 MGC803 GES-1 CCD841	Induced apoptosis associated with decreased FAO, NADPH/NADP+ ratio, and mitochondrial transmembrane potential. Increased ROS levels. Synergistic with oxaliplatin.
Zhu et al. (2019) [33]	EOC	OVCAR3, CAOV3, OV90	Inhibited Akt/mTOR/S6K. Increased apoptosis
Kant et al. (2020) [34]	GBM	PN19 MES83 T98G U251	Anti-tumoral effects of PHX were independent of CPT and FAO inhibition.
Agren et al. (2014) [35]	HCC	HepG2	Inhibited growth.
Brown et al. (2018) [36]	HCC	Murine and human CD4+ T cells	Rescued fatty acid-induced apoptosis.
Xu et al. (2018) [37]	HCC	Hep3B Huh7	Showed effects on glycolysis, OXPHOS and FAO. Inhibited growth. Induced apoptosis. Upregulated AMPK.
Xu et al. (2019) [38]	MM	RPMI8226 OPM2	Decreased viability. Induced apoptosis.
Vella et al. (2015) [39]	NB		Increased expression of NDM29 ncRNA Downregulated ABC transporter (*ABCA1, ABCA12*) and solute carrier (*SLC7A11*) expression. Synergistic with cisplatin.
Rathore et al. (2021) [40]	OSS	NOS1	Did not alter oxygen consumption. Inhibited cell proliferation, induced cell deat, and reduced total RSP6 and mTOR at higher concentration. Synergistic with NCT-503.
Ghaffari et al. (2015) [41]	PC, SCC	PC-3, A-431	Reduced viability.
Flaig et al. (2017) [42]	PC	22Rv1 MDV3100-resistant LNCaP TRAMPC1	Combination treatments synergistically reduced proliferation.
Itkonen et al. (2017) [43]	PC	LNCaP	Increased intracellular lipid accumulation. Decreased proliferation. Induced apoptosis and incomplete autophagy. Blocked proliferation in combination with MVD-3100 or ABI.
Nassar et al. (2020) [44]	PC	LNCaP C4-2B 22RV1	Decreased viability of cells. Downregulated expression of cell-cycle related genes CDK4, CDK6, AURKB, CCD20, CCND1, CCNE2, and E2F1 Increased G0–G1 cells. Increased cleaved PARP levels and apoptotic cells. Synergistic with AUY922.
Schnell et al. (2015) [45]	T-ALL	HPB-ALL DND41 JURKAT CCRF-CEM CUTLL1 RPMI8402	Induced strong anti-leukemic responses in T-ALL cells with and without NOTCH1 mutations. Anti-leukemic in primary human T-ALL.

**Table 2 molecules-28-03624-t002:** Effects of perhexiline treatment in preclinical murine cancer models.

Study	Cancer	Mouse Strain	Model	Treatment	Key Findings
Ren et al. (2015) [24]	BRCA	SCID	MDA-MB-468 xenograft, s.c.	Monotherapy PHX 400 mg/kg, intragastric, 5 days/week, 4 weeks.	PHX significantly inhibited tumour growth, and decreased HER3 activation (pHER3).
Liu et al. (2016). [30]	CLL	Tcl-1Tg: p53^−/−^ transgenic	Spontaneous CLL	Monotherapy PHX 8 mg/kg, i.p., every other day for 4 injections.	PHX selectively eliminated CLL cells, significantly reduced leukemic burden and prolonged OS.
Wang et al. (2020) [32]	CRC	BALB/c nude	HCT116, xenograft, s.c. dorsal flank	Monotherapy and combination therapy; CDDP 5 mg/kg, once/week, 4 weeks; PHX 8 mg/kg, every second day, 4 weeks.	PHX monotherapy, and PHX and CDDP combination therapy reduced tumour progression. PHX and CDDP combination overcame resistance in CDDP-resistant cell line (HCT116/OXA).
	CRC/GC	NSG	PDX, s.c., dorsal flank	Monotherapy and combination therapy; CDDP 5 mg/kg, once/week, 4 weeks; PHX 8 mg/kg, every second day, 4 weeks.	PHX and CDDP monotherapy and combination therapy inhibited proliferation (Ki-67) and increased apoptosis (TUNEL).
	GC	BALB/c nude	HGC27 xenograft, s.c. dorsal flank	Monotherapy and combination therapy; CDDP 5 mg/kg, once/week, 4 weeks; PHX 8 mg/kg, every second day, 4 weeks.	PHX monotherapy, and PHX and CDDP combination therapy reduced tumour progression.
Kant et al. (2020) [34]	GBM	Nu/Nu nude	MES83 xenograft, s.c. (flank) and orthotopic (brain).	PHX monotherapy, 80 mg/kg, intragastric, 5 days/week, up to 24 days.	PHX accumulated in the brain. PHX significantly reduced growth of flank and orthotopic tumours, and increased overall survival.
Xu et al. (2018) [37]	HCC	Nu/nu nude	Hep3B, Huh7, and HepG2 xenograft, s.c.; H460 isogenic lung, s.c.	Triple combination; PHX 30 mg/kg, i.p., daily; DPI 2 mg/kg, i.p., daily.	Triple combination of HK2 knockdown, DPI and PHX significantly inhibited tumour growth, increased apoptosis, decreased AMPKα and phosphorylation of S6.
Brown et al. (2018) [36]	HCC	Liver specific inducible MYC oncogene (MYC-ON)	Spontaneous HCC	Monotherapy; PHX 8 mg/kg, i.p., 3/week, 5 weeks.	PHX decreased incidence of HCC in NAFLD model. PHX reduced early apoptotic events in intrahepatic CD4+ T cells.
Xu et al. (2019) [38]	MM	NSG	OPM-2 (HK1^−^HK2^+^) and U266 (HK1+HK2^+^) xenografts, s.c., and P3X63Ag (HK1^−^HK2^+^), s.c.	Triple combination therapy; PHX 30 mg/kg i.p. daily; HK2-ASO1 50 mg/kg, s.c.; DPI 2 mg/kg or MET 250 mg/kg, i.p. daily.	Triple combination of HK2-ASO1, DPI or MET, and PHX significantly inhibited tumour progression, and increased PARP-1 cleavage in OPM-2 (HK1^−^HK2^+^), but not U266 (HK1+HK2^+^) xenografts. Triple combination of murine HK2-ASO1, DPI or MET, and PHX significantly inhibited tumour progression in P3X63Ag (HK1^−^HK2^+^) murine MM cells, and prolonged OS.
Vella et al. (2015) [39]	NB	NOD-SCID (NOD.CB17-Prkdscid)	SK-N-BE(2) xenograft, s.c.	Monotherapy and combination therapy; PHX 1 or 3 mg/kg, intragastric, 5 days/week; CDDP 3 or 5 mg/kg, i.p., once/week.	PHX monotherapy (1 or 3 mg/kg/dose) did not alter tumour growth. PHX (1 mg/kg) and cisplatin (3 mg/kg) combination reduced tumour growth. PHX (3 mg/kg) and cisplatin (5 mg/kg) combination reduced tumour growth, significantly increased progression-free survival, and inhibited cisplatin-induced increase in the NB cell differentiation marker, neurofilament 68 (NF68).
Rathore et al. (2021) [40]	OSS	Athymic nude	U2OS xenograft, s.c.	Monotherapy or combination therapy; PHX 8 mg/kg, intragastric, daily for 30 days; NCT-503 40 mg/kg, i.p., daily for 30 days.	PHX monotherapy, but not NCT-503, moderately reduced tumour progression. PHX and NCT-503 combination therapy markedly reduced tumour progression resulting in sustained inhibition over 30 days.
Zhu et al. (2019) [33]	EOC	BALB/c nude	OVCAR (*NKX2-8*^+/−^) xenograft, i.p.	Monotherapy or combination therapy; CDDP 5 mg/kg every 3 days; PHX 3 mg/kg.	PHX and CDDP combination therapy markedly reduced tumour progression resulting in sustained inhibition over 6 weeks, prolonged OS and induced apoptosis (TUNEL and activated caspase 3).
Schnell et al. (2015) [45]	T-ALL	C57BL/6	NOTCH1-induced murine T-ALL	Monotherapy; PHX 53.68 mg/kg.	PHX reduced tumour burden (bone marrow cellularity and leukaemic infiltration, spleen weight and cellularity), and increased OS.

## Data Availability

Not applicable.

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
