# Peer review of "Perhexiline: Old Drug, New Tricks? A Summary of Its Anti-Cancer Effects"

_molecules, 2023, doi:10.3390/molecules28083624_

Round 1
Reviewer 1 Report
This is a very nice and complete review about perhexiline use as an anti-cancer agent. There are some minor spell errors that must be corrected. For example, on the line 423 it is written that "perhexiline has the potential to potentiate..." This sounds odd and I suggest it to be re-written.
Reviewer 2 Report
In this current manuscript, Dhakal et al summarized the significance of targeting altered fatty acid metabolites in cancer therapy by the known anti-angina drug perhexiline. Alter Fatty acid metabolism is one of the crucial metabolic adaptations of cancer stem cells in a hypoxic microenvironment and also provides resistance to conventional chemo and immune therapy. Hence, studies targeting these pathways can have wide implications in future cancer therapy. Here, the authors have done a thorough review of literate study and summarized the improvement made in this field. Over all, this is an interesting review to read, and gives a broader view of this field. The manuscript was written nicely and summarized the major studies done in this field. This review will probably get future attention from a wide range of basic and clinical cancer researchers.
Comments- No major comments.
Minor-
1. The author can separately add a paragraph for any limitation and challenge and future direction.
2. How perhexiline modulates tumor immune components, cancer immune therapy effect and targets cancer stem cells ….. A little evidence will improve the quantity and quality of the review.
3. Table 1 heading needs to be changed. Like…”Studies summarizing the effect of Perhexiline on various cancer cells”
4. Fig. 1 schematic diagram is very simplified. It should be improved.
5. Rewrite line 50 for reference 8.
Reviewer 3 Report
1. perhexiline is an inhibitor of CPT. Please briefly describe the recommended advantages of perhexiline over other CPT inhibitors for clinical use.
2. If the authors improve Figure 1 and add that perhexiline exerts its biological effects by influencing which proteins expression, readers may understand the mechanism of perhexiline more intuitively.
3. In Part 4, are the several biological target proteins independent of the CPT inhibition pathway directly or indirectly targeted, or are they mediated by other intracellular components? Please clarify.
4. The full text does not use abbreviation ETO (Line485), please check.
Reviewer 4 Report
Dear author,
The paper is a good, and well-supported proposal related to the repositioning of drugs in the cancer treatment. I only recommend:
1. Change the title.
2. Change the format of figure 1 because it is very simple.
3. In section 6. Clinical feasibility of perhexiline as an anti-cancer agent, The author must make an integrative figure with the potential signaling pathways affected.
Kind regards,
